# Spider Silk Fibroin Protein Heterologously Produced in Rice Seeds Reduce Diabetes and Hypercholesterolemia in Mice

**DOI:** 10.3390/plants9101282

**Published:** 2020-09-28

**Authors:** Won Tae Yang, Kwang Sik Lee, Yeon Jae Hur, Bo Yeon Kim, Jianhong Li, Sibin Yu, Byung Rae Jin, Doh Hoon Kim

**Affiliations:** 1College of Life Science and Natural Resources, Dong-A University, Busan 49315, Korea; wtyang@dau.ac.kr (W.T.Y.); leeks@dau.ac.kr (K.S.L.); my48642@hanmail.net (Y.J.H.); boyeon@dau.ac.kr (B.Y.K.); 2College of Plant Science and Technology, Huazhong Agricultural University, Wuhan 430070, China; jianhl@mail.hzau.edu.cn; 3National Key Laboratory of Crop Genetic Improvement, College of Plant Science and Technology, Huazhong Agricultural University, Wuhan 430070, China; ysb@mail.hzau.edu.cn

**Keywords:** transgenic rice, spider silk fibroin protein, glucose metabolism, lipid metabolism

## Abstract

Silk fibroin proteins are biomaterials with diverse applications. These spider and silkworm proteins have specific biological effects when consumed by mammals; in addition to reducing blood pressure and blood glucose and cholesterol levels, they have anti-human immunodeficiency virus activity. In the present study, rice (*Oryza sativa*) was engineered to produce the C-terminus of the major ampullate spidroin protein from the spider *Araneus ventricosus* under the control of a *Prolamin* promoter. Homozygous transgenic rice lines were identified, and the therapeutic effect of this spider silk fibroin protein on the lipid and glucose metabolism was analyzed in a mouse model. Feeding fat-fed mice, the transgenic rice seeds for four weeks reduced serum concentrations of triglycerides, total cholesterol, low-density lipoprotein cholesterol, glutamic oxaloacetic transaminase, and glutamic pyruvic transaminase, and lowered blood glucose levels. This is the first study to investigate the effects of consumption of rice seeds heterologously expressing spider silk fibroin protein in a mammalian model. Our findings suggest that functional foods containing spider silk fibroin protein might be useful as potential pharmaceutical materials for preventing and treating diabetes, hyperlipidemia, and hypercholesterolemia

## 1. Introduction

Silk fibroin proteins from spiders or silkworms are attractive biomaterials for biotechnological applications because of their biocompatibility, biodegradability, and mechanical properties [1,2,3,4]. They have been used in a wide range of products, including biomedicines, textiles, personal hygiene products, and cosmetics; for instance, silk fibroin proteins have been used in biopolymers, contact lens materials, biomaterial membranes, microcapsules, and bioengineered bone grafts [1,2,3,5]. Silk fibroin proteins are mainly composed of glycine (G) and alanine (A) residues. Among the spider silks, the major ampullate spidroin proteins contain ubiquitous poly-A stretches and GGX (X = A, Q, or Y) peptide motifs [6,7]. The heavy chain fibroin silk from silkworms contains 2377 repeats of a GX dipeptide motif, where the X residues are A (in 64% of the repeats), S (22%), Y (10%), V (3%), and T (1.3%) [8].

Silk fibroin proteins have also been reported to have therapeutic effects when consumed by mammals, including the enhancement of glucose [9,10,11,12,13,14] and lipid [12] metabolism, anti-HIV activity [15], DNA damage protection [11], and the reduction of blood pressure [16]. Diabetes and hypercholesterolemia are key causes of death in the global adult population. Since rice (*Oryza sativa*) is the staple food for a majority of the world’s population, introducing silk fibroin proteins into transgenic rice plants is an attractive approach for potentially improving human health [17]. Rice seeds, in contrast to many other plant tissues, can accumulate high levels of recombinant proteins and facilitate their long-term stable storage [18]. 

We previously reported the cloning and expression of partial cDNAs encoding the C-terminus of the spider (*Araneus ventricosus*) major ampullate silk protein (AvMaSp), which was found to enhance insulin secretion and reduce blood glucose levels when consumed by diabetic mice [7,14]. We found that the recombinant AvMaSp spider silk fibroin protein exhibits a non-cytotoxic and non-inflammatory response [19]. Although spider silk fibroin proteins have been studied for their anti-diabetic effects in type 2 diabetic mice, the biological effects of spider silk protein produced in transgenic rice seeds on processes such as glucose and lipid metabolism remain unexplored.

In this study, we aimed to develop a transgenic rice plant potentially accumulating high levels of glycine and alanine derived from AvMaSp in the seeds, and to explore its therapeutic potential. To achieve this, we introduced a spider silk protein gene encoding the C-terminus of AvMaSp into rice, generating a homozygous transgenic rice line that accumulates AvMaSp in the seeds. The oral administration of the seeds reduced blood glucose, and cholesterol levels in mice. This is the first report to use a direct oral administration of rice seed-produced silk fibroin protein to effectively treat diabetes and hypercholesterolemia in mice.

## 2. Results 

### 2.1. Generation and Characterization of a Transgenic Rice Line

To generate the transgenic rice plants, we used a pCAMBIA1300 vector containing the *AvMaSp* gene driven by the *Prolamin* promoter, a Tnos terminator, and a hygromycin-resistant selection marker (Figure 1A). The *AvMaSp* vector was introduced into rice using an *Agrobacterium*-mediated system. To verify the genotype of the transgenic lines, *AvMaSp* was amplified from genomic DNA using gene-specific primers, resulting in the amplification of a 795-bp *AvMaSps* sequence from the transgenic lines but no amplification from the wild type (Dong-Jin cultivar; Figure 1B). Southern blot analysis verified that a single copy of *AvMaSp* had been inserted into two of the transgenic lines (line 3 and 5) (Figure 1C). Using genome walking, we identified the location of *AvMaSp* in the genomes of these lines. An analysis of the flanking DNA sequence confirmed that *AvMaSp* was located in chromosomes 10 and 3 of lines 3 and 5, respectively (Figure 1D). 

Total RNAs and proteins were extracted every 10 days after the flowering stage until the seed maturation stage (Figure 2A). To determine the expression level of *AvMaSp*, we performed an RNA blot analysis of the transgenic rice seeds of the homozygous T_3_ generation of lines 3 and 5, which were selected except for the transgenic rice with low gene expression and growth rate (Appendix A). The *AvMaSp* expression increased from 20 to 40 days after flowering in both transgenic rice lines, while it was not detected at any stage in the Dong-Jin seeds (Figure 2B). Immunoblot analysis using an anti-AvMaSp antibody showed that AvMaSp protein was present in the transgenic seeds between 20 and 40 days after flowering (Figure 2C and Appendix A). 

Next, we examined the localization of AvMaSp in the transgenic seeds using immunofluorescence staining. AvMaSp was distributed throughout the embryos of the transgenic lines, but was absent in the wild-type seeds (Figure 3). Thus, *AvMaSp* is expressed in the transgenic rice seeds and particularly the protein accumulated in the embryo.

### 2.2. Effects of the Transgenic Rice on Mouse Liver Weights and Blood Glucose Levels

To determine whether the consumption of transgenic rice expressing *AvMaSp* can decrease the body and liver weights of mice, we fed either a standard diet, a wild-type rice-containing diet, or a transgenic rice-containing diet to the mice for four weeks. During the experimental period, the diet containing the transgenic rice expressing *AvMaSp* did not induce changes in the body weights of either the C, FC, DC or TR groups, whereas the liver weights were significantly lower in the TRL3-23 (*p* = 0.00621) group than in the DC-23, and TRL3-46 (*p* = 0.017246) and TRL5-46 (*p* = 0.011074) groups than in the DC-46 group (Appendix A). 

The blood glucose levels were also measured in the mice after four weeks of consuming these experimental diets. The FC group had a higher blood glucose level of approximately 170 mg/dL, whereas the other groups had significantly lower blood glucose levels (70–110 mg/dL), indicating that they were maintained at normal levels (Figure 4). Especially, the TRL3-23 and TRL5-23 group showed the reduced blood glucose levels compared to DC-23, but not in the TRL3-46 and TRL5-46 group. These results suggest that the liver weights and blood glucose levels of mice might be influenced by the consumption of the transgenic rice expressing *AvMaSp.*

### 2.3. Effects of the Transgenic Rice on the Mouse Lipid Metabolism 

The levels of triglycerides and cholesterol in the mouse sera were analyzed to evaluate the functional effects of the diet containing transgenic rice expressing *AvMaSp*. The triglyceride levels of the FC and DC groups were significantly higher than those of the C, TRL3-23, TRL3-46, and TRL5-46 groups except the TRL5-23 group (Figure 5A). After four weeks of treatment, the TRL groups showed the significant reduced levels of total cholesterol and LDL cholesterol compared to the DC groups (Figure 5B,C). The levels of total cholesterol in the TRL3-23 and TRL5-23 groups were reduced with levels 18% and 17% than that of the DC-23 group, and the TRL5-46 group reduced with 28% than that of the DC-46 group. The TRL3-23 group showed the reduced level of LDL cholesterol with 49% compared to the DC-23 group and the TRL5-46 group particularly substantially reduced, with level 71% lower, respectively, than those of the DC-46 group. The TRL3 treatment did not induce changes in the HDL cholesterol levels, but TRL5 treatment induced changes in that compared to DC treatments (Figure 5D).

### 2.4. Effects of the Transgenic Rice on Mouse Liver Function

To examine the biochemical effects of the transgenic rice when consumed by mice, we determined the levels of the glutamic oxaloacetic transaminase (GOT) and glutamic pyruvic transaminase (GPT) enzymes in the mouse sera. The levels of GOT and GPT in the FC group were significantly increased, while the TRL3 and TRL5 groups showed markedly reduced levels of GOT and GPT, indicating that consuming the transgenic rice expressing *AvMaSp* could effectively improve liver function (Figure 6A,B).

## 3. Discussion

Silk fibroin proteins from silkworms and spiders elicit very low levels of macrophage stimulation and are non-cytotoxicity, non-inflammatory, and exhibit low antigenicity when consumed by mammals [19,20,21]. They are well-known to have pharmacological activities, including anti-diabetic [10,11,12,13,14,22] and anti-obesity [23,24,25,26] properties. Silkworm and spider silk fibroin proteins enhance insulin secretion and reduce blood glucose levels [12,19]. In addition, silkworm and spider silk proteins are rich in glycine and alanine residues [1,8,27,28,29]. 

One way to address the healthcare issues of people around the world is through the enhancement of staple crops to increase their essential nutrient content [30] or to introduce health-promoting properties via genetic engineering [17]. Golden Rice is a promising example of a transgenic rice plant that has been engineered to provide tangible benefits to meet social and economic [31,32]. Recently, various additional attempts have been made to enhance the health-promoting properties of rice plants, including the development of transgenic rice for allergy immunotherapy [33], the engineering of transgenic rice seeds to produce a type II-collagen (CII) tolerogenic peptide to tackle anti-CII autoimmune diseases [34] or human insulin-like growth factor 1 for treating diabetes [35], and the development of gamma-aminobutyric acid (GABA)-enriched rice to reduce hypertension [36].

In this study, we constructed transgenic rice plants that produce AvMaSp, which is rich in glycine and alanine residues. Two lines were selected for further analysis, based on their genotype and protein contents. To evaluate the functional effects of these glycine and alanine-enriched rice grains, mice were fed a diet containing either of the transgenic lines or Dong-Jin (wild-type) rice for four weeks, with a standard diet used as the control. The supplementation of the transgenic rice seeds reduced the liver weights and blood glucose levels in the mice to levels similar to those observed in the control group. Glycine was previously reported to affect insulin secretion and reduce blood glucose levels [14,37,38,39,40]. Therefore, the high glycine content of the fibroin silk protein may be a key factor for this reduction of blood glucose levels in mice. Feeding fat-fed mice the transgenic rice seeds also reduced their total lipids, total cholesterol, and low-density lipoprotein cholesterol levels. In conclusion, although our results were performed under the limited conditions such as 5 mice and 4 weeks of treatment in each group, our findings show that engineering rice to produce a spider silk fibroin protein could have potential pharmaceutical applications as preliminary data to provide an alternative food-delivered therapeutic for diabetes, hyperlipidemia, and hypercholesterolemia.

## 4. Materials and Methods

### 4.1. Production of Transgenic Rice

The cDNA encoding a partial repetitive region and the C-terminal domain of AvMaSp [8] was PCR-amplified from *pBluescript-AvMaSp* using the forward primer (1–24), 5′-GGTACCATGGCAGCAGCTGCAGCAGCAGCAGCCGG-3′ (*Kpn*I site introduced) and reverse primer (775–795) 5′-GAGCTCTTAAGAAAGAGCTTGGTAAAC-3′ (*Sac*I site introduced). The isolated *AvMaSp* fragment was inserted into the binary pCAMBIA1300 vector (*pCAMBIA-AvMaSp*) containing the prolamin promoter, which is a seed specific promoter, and Tnos terminator. To make transgenic rice containing the partial AvMaSp gene, pCAMBIA-AvMaSp was introduced into *Agrobacterium tumefaciens* (EHA105) by electroporation. We used a modified version of a general rice transformation protocol [41,42].

### 4.2. DNA and RNA Gel Blot Analysis

Total genomic DNA was isolated from rice leaves using the cetyltrimethyl ammonium bromide extraction method [43]. A total of 5 µg of genomic DNA digested with appropriate enzyme were separated by electrophoresis on 1% (*w*/*v*) agarose gels and blotted on nylon transfer membranes (Amersham). Total RNA was isolated from rice seeds using the TRIzol reagent. Total RNA, 10 µg, was separated by electrophoresis on 1.2% (*w*/*v*) denaturing formaldehyde agarose gels and blotted on nylon transfer membranes. The membranes were hybridized at 65 °C for 12 h with [α-^32^P]-dCTP-labeled probe in a hybridization Church buffer. The membranes were washed twice in 2 × SSC and 0.2% (*w*/*v*) SDS at 65 °C for 10 min, twice in 1 × SSC and 0.2% (*w*/*v*) SDS at 65 °C for 15 min, and twice in 0.1 × SSC and 0.2% (*w*/*v*) SDS at 65 °C for 20 min before autoradiography.

### 4.3. Genome Analysis

To detect the location of the silk fibroin protein gene into transgenic rice genome, use DNA working SpeedUp^TM^ premix kit (Seegene) to obtain flanking sequences that lie adjacent to the silk fibroin protein gene. A PCR fragment of the expected band was purified from agarose gel, followed by cloning into the plasmid pGemT-easy vector (promega). The insert DNA was sequenced commercially. A search using the phrase flanking sequences was conducted at the Gramene and NCBI. Matched sequences were found significant at E values 10-5 for FSTs, BLASTN searches. 

### 4.4. Western Blot Analysis

To explore expression of transgene during seed maturation at the protein level, transgenic rice seeds were harvested every 10 days between 0 and 40 days after the flowering stage. Total protein was isolated from rice seeds using PBS buffer and total protein of transgenic plants, all the samples were resolved on SDS-PAGE and transferred to nitrocellulose membrane. The nitrocellulose membrane was incubated in blocking buffer (1% bovine serum albumin) for 1 h at room temperature, with antiserum of AvMaSp (Lee et al., 2014) for an appropriate time, and the with anti-mouse IgG horseradish peroxidase (HRP) conjugate secondary antibody for another 1 h. The protein levels were visualized with the ECL plus western blotting detection system (Amersham) after extensive washing with washing buffer.

### 4.5. Immunofluorescence Staining

Transgenic rice seed was fixed for 12 h with 4% neutral buffered paraformaldehyde at room temperature. Tissue was embedded in optimal cutting temperature compound (Miles Scientific, Naperville, IL, USA) and stored at −72 °C. Sections were washed 3 times with PBS, once with PBS containing 2 mg/mL BSA and incubated at room temperature with a 300 times dilution of antiserum of silk fibroin protein gene. The slides were washed 3 times with PBS and goat anti-mouse IgG conjugated to FITC and extensively washed in PBS. The slides were then mounted in 1,4-Diazabicyclo-octane (Sigma, St. Louis, MO, USA) and observed under a Leica TCS 4D connected to an Olympus 1 × 70 upright microscope (Olympus, Tokyo, Japan) equipped for fluorescence.

### 4.6. Animals

Balb/c male mice were purchased at 4 weeks of age from Samtako Bio Korea Co. (Samtako Bio Korea, Osan, Korea). The experimental mice were kept at 22 ± 2 °C under a 12-h light/dark cycle with a relative humidity of 45–55%, as previously described [15]. After a week for an adaptation, the mice were divided into eight groups (5 mice/group) according to the treatment received: the C group (fed with a standard laboratory diet), FC group (fed with 7% lard), DC-23 group (fed with 7% lard and 23% Dong-Jin rice), DC-46 group (fed with 7% lard and 46% Dong-Jin rice), TRL3-23 group (fed with 7% lard and 23% line 3 of transgenic rice), TRL3-46 group (fed with 7% lard and 46% line 3 of transgenic rice), TRL5-23 group (fed with 7% lard and 23% line 5 of transgenic rice), and TRL5-46 group (fed with 7% lard 46% line 5 of transgenic rice). The ingredient composition of the diet is presented in Appendix A. The groups were fed for 4 weeks with 2 g of the diet per a day. After 4 weeks, body weights of all mice were individually measured with an animal balance (Mettler Toledo, Lutz, FL, USA). The experimental protocol for the animals was approved by the Dong-A University Animal Care Committee (approval no. DIACUC-13-20).

### 4.7. Sample Collections

Mice were sacrificed on 4 weeks to collect liver tissue samples and blood. Liver samples were immediately washed with phosphate-buffered saline (PBS; 140 mM NaCl, 27 mM KCl, 8 mM Na_2_HPO_4_, and 1.5 mM KH_2_PO_4_, pH 7.4), cleaned using paper towels to remove liquid and then the liver weight were obtained. Blood samples were collected from the heart using a heparinized syringe and the plasma was separated by centrifugation at 3000× *g* for 15 min at 4 °C, and then stored at −70 °C until use.

### 4.8. Analyses

Glucose levels were measured using a Glucose analyzer (Glucotrend, Roche, Mannheim, Germany) according to the manufacturer’s protocol. The levels of plasma triglyceride were analyzed using a Triglyceride Quantification Colorimetric Kit (BioVision Inc., Milipitas, California, USA) according to the manufacturer’s instructions. The levels of cholesterols in the plasma were quantified by Total Cholesterol and Cholesteryl Ester Colorimetric Assay Kit (BioVision Inc., Milpitas, CA, USA) according to the manufacturer’s instructions. The levels of aspartate aminotransferase (GOT) and glutamic pyruvic transaminase (GPT) in serum were evaluated using an Alanine Aminotransferase (ALT or SGPT) Activity Colorimetric Assay Kit (BioVision Inc., Milpitas, CA, USA) and Asparate Aminotransferase (AST or SGOT) Activity Colorimetric Assay Kit (BioVision Inc., Milpitas, CA, USA) according to the manufacturer’s instructions.

### 4.9. Statistical Analysis

Data are shown as the means ± SDs. The data were analyzed using an independent unpaired 2-tailed Student’s t-test. All statistical analyses were performed using SPSS PASW 22.0 package for Windows (IBM, Chicago, IL, USA). The statistical significance was set to ** *p* < 0.01 and * *p* < 0.05.

## Figures and Tables

**Figure 1 plants-09-01282-f001:**
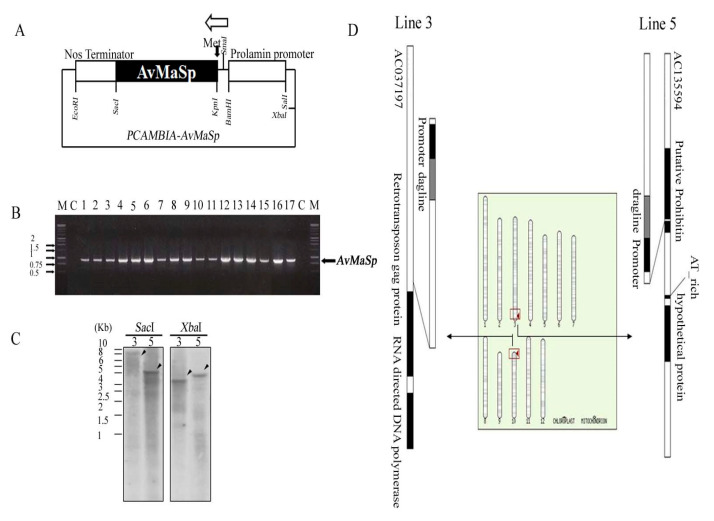
Characterization of the transgenic rice lines. (**A**) Constructed *pCAMBIA-AvMaSp* vector. To generate the transgenic rice plants, *AvMaSp* was introduced under the control of the *Prolamin* promoter. (**B**) Screening of the transgenic lines expressing *AvMaSp* using PCR. The *AvMaSp* gene was amplified from the genomic DNA of the transgenic rice lines using gene-specific primers, with Dong-Jin DNA used as the control C. (**C**) Identification of the single-copy *AvMaSp* insertion in the transgenic lines using a Southern blot analysis. (**D**) Determination of the location of the *AvMaSp* gene in the rice genome using a flanking DNA sequence analysis.

**Figure 2 plants-09-01282-f002:**
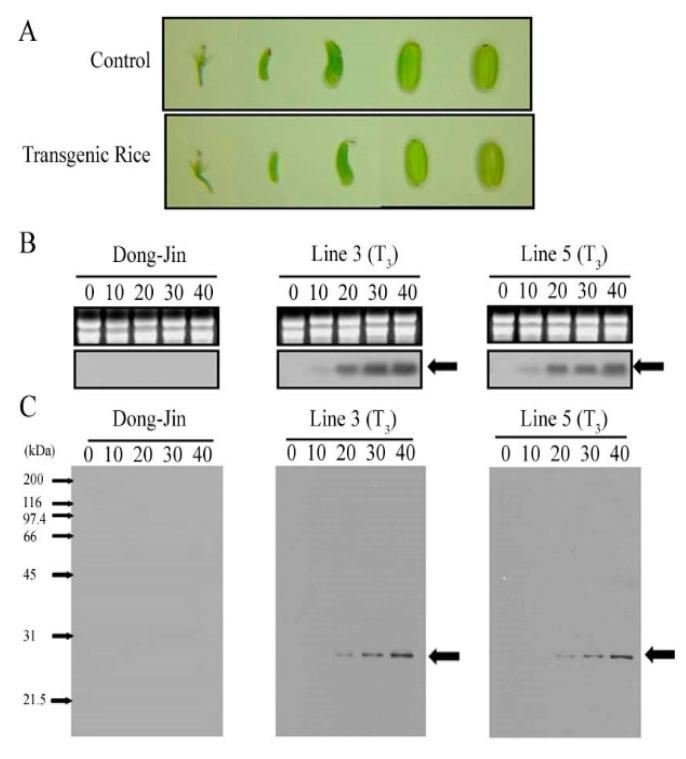
*AvMaSp* expression profiles during the development of the transgenic seeds. (**A**) Development of the transgenic seeds. The seeds were imaged every 10 days between the flowering stage and seed maturation. (**B**) Transcriptional expression profiles of *AvMaSp* in the transgenic lines, revealed using an RNA blot. Total RNA was extracted from the seeds harvested in (**A**). *AvMaSp* transcripts are indicated with an arrow. The ethidium bromide-stained RNA gel shows uniform loading (upper panel). (**C**) Translational expression profiles of the AvMaSp protein in the transgenic lines, revealed using a western blot. The proteins were extracted from the seeds harvested in (**A**). The AvMaSp proteins are indicated with an arrow.

**Figure 3 plants-09-01282-f003:**
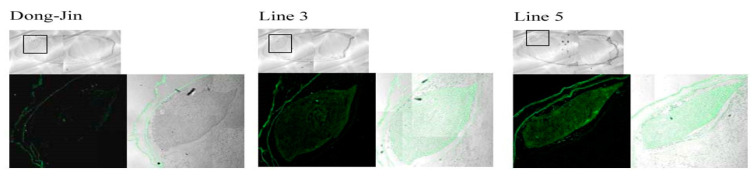
Immunofluorescence staining of AvMaSp in transgenic rice seeds. AvMaSp was found to be localized in the region of the embryo in the transgenic lines. Dong-Jin was used as the wild-type control.

**Figure 4 plants-09-01282-f004:**
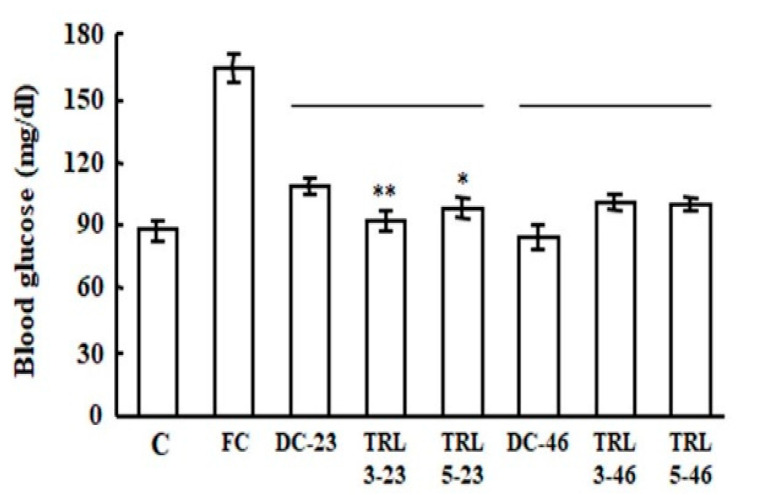
Effects of the transgenic rice on blood glucose levels. The blood glucose levels were quantified after four weeks of the oral administration of the various experimental diets. The statistical significance was set to ** *p* < 0.01 and * *p* < 0.05

**Figure 5 plants-09-01282-f005:**
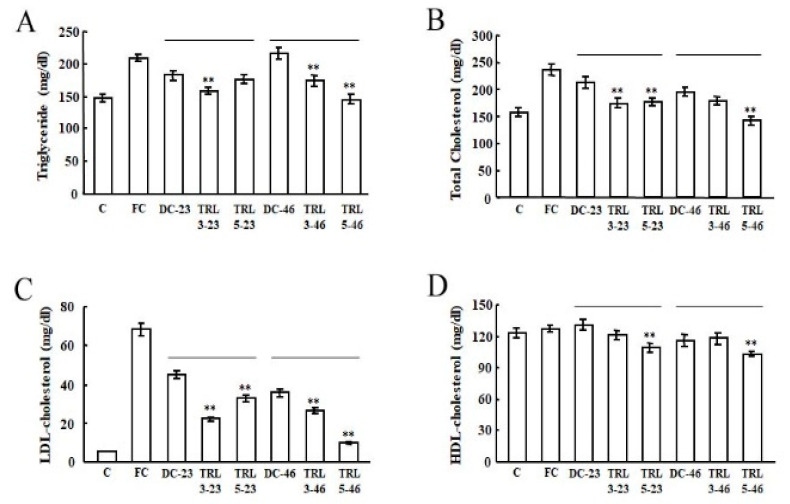
Effects of the transgenic rice on the lipid metabolism in the mouse sera. The levels of triglycerides (**A**), total cholesterol (**B**), high-density lipoprotein (HDL) cholesterol (**C**), and low-density lipoprotein (LDL) cholesterol (**D**) in the mouse sera were quantified after four weeks of the oral administration of the various experimental diets. The statistical significance was set to ** *p* < 0.01.

**Figure 6 plants-09-01282-f006:**
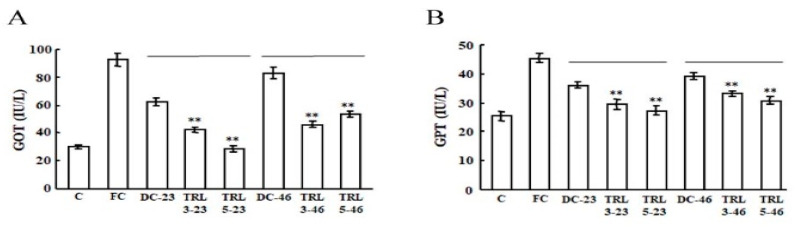
Effects of the transgenic rice on the liver function of mice. The levels of glutamic oxaloacetic transaminase (GOT; **A**) and glutamic pyruvic transaminase (GPT; **B**) in the mouse sera after four weeks of the oral administration of the various experimental diets. The statistical significance was set to ** *p* < 0.01.

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
