# Peer review of "Spider Silk Fibroin Protein Heterologously Produced in Rice Seeds Reduce Diabetes and Hypercholesterolemia in Mice"

_plants, 2020, doi:10.3390/plants9101282_

Round 1
Reviewer 1 Report
- Authors are encouraged to show the restriction enzyme site XbaI on the plasmid map.
- Authors are encouraged to show SDS PAGE gel for western blot, picture on the supplementary files
- Is there any morphological difference between the transgenic lines? Authors are encouraged to show picture of transgenic rice lines.
- What are the statistical methods followed to depict the results?
- Line 200 , typo error, should be changed to flanking sequence..
- Line 218 , please expand the OCT, abbreviate it..
- Authors are encouraged to abbreviate the new term wherever possible
- Did authors mention anywhere, why did the transgenic line 3 and 5 were chosen for the animal study, if not, please mention
Reviewer 2 Report
In their paper, Yang et al, describe the construction of transgenic rice lines expressing the major ampullate spidroin protein from the spider Araneus ventricosus under the control of a Prolamin promoter. By using diverse molecular techniques, they prove the successful construction of two rice lines and they subsequently feed laboratory mice with standard diets and diets that contain lard and transgenic rice seeds (in various concentrations). They find that triglycerids, total cholesterol, LDL cholesterol, GOT and GPT are all reduced in mice fed with transgenic rice along with blood glucose levels. Although this is an interesting finding showing a significant potential for the use of transgenic rice in human diets, I believe that the Introduction and Discussion sections of the article are not very extensive. They are both hardly covering a page. Perhaps the authors should enhance a bit more these two sections; for example, by including more similar works in the discussion section that would make their findings more relevant and strengthen the paper. Therefore, and for the specific comments cited below, I believe the paper could be reconsidered after major revisions.
Specific comments;
- My major concern is in section 2.3 on the effects of the transgenic rice on the mouse lipid metabolism where the authors mention that differences are significantly higher or lower among treatments. However, they do not mention which tests of statistical significance or statistics they used in their Materials and Methods section? Are the differences statistically significant between treatments? I can see some statistical analysis in Supplementary Table 1 when it comes to body weight and liver weight but no further explanation.
- Also, could the authors elaborate a bit more on why they used these specific contents in mice diets e.g. 7% lard and 23-46% transgenic rice (Supplementary Table 1)? Is there a reference study that shows that these are the appropriate diet compositions they should use?
- Authors indicate they used a 795bp AVMaSp fragment which contains “the cDNA encoding a partial repetitive region and the C-terminal domain of AvMaSp”. Although they do refer to a previous paper published in 2012, can they explain if they included the whole gene in their constructs?
- Why section 2 is mentioned as the Results and Discussion section when there is a separate Discussion section (section 3)? I believe the journal is not accepting a combined Results and Discussion section.
- Lines 129-130; Throughout the experimental period, the TRL treatments did not induce changes in the HDL cholesterol levels – that’s in Figure 5C not in Figure 5D.
Reviewer 3 Report
Dear authors
my comments are in the attached document.

Round 2
Reviewer 2 Report
The manuscript has been improved with the addition of supplementary figures and the statistical analyses. I appreciate authors’ efforts to address my comments at the first round of reviews but there are still some issues to be addressed;
Lines 75-77; the authors have inserted a line here to explain why they selected transgenic lines 3 and 5 for further analysis. They say “…. which were selected except for the transgenic rice with low gene expression and growth rate (Fig. S1)”. However, in the northern blot of Fig. S1 where AvMasp overexpression is depicted in the transgenic To lines, I can see more lines that overexpress the transgene. Could the authors explain a bit more what do they mean by selecting lines 3 and 5 for their growth rate?
In Figure 4; Authors mention that The FC group had a higher blood glucose level of approximately 170 mg/dL, whereas the other groups had significantly lower blood glucose levels (70~110 mg/dL), indicating that they were maintained at normal levels (Fig. 4). Taken together, these results suggest that the liver weights and blood glucose levels of mice were influenced by the consumption of the transgenic rice expressing AvMaSp.” The statistical analysis shows that the FC group fed with 7% lard has indeed higher blood glucose levels. However, the rest of the treatment groups both the ones fed with normal rice and transgenic rice have the same blood glucose levels as the control. So why do the authors attribute this decrease only to transgenic rice expressing AvMaSp when the groups fed with Dong-Jin rice have approx. the same blood glucose levels?
Lines 127-128; C, TRL3 and TRL5 groups had lower levels of total cholesterol and LDL in comparison to what? To FC group?
Lines 130-131; “Throughout the experimental period, the TRL treatments did not induce changes in the HDL cholesterol levels (Fig. 5D).” I can see some statistically significant differences in TRL 5-23 and TRL5-46, why do the authors say that there are no changes when there’s a slight but statistically significant drop in HDL?
Reviewer 3 Report
Dear authors
I added some more comments and questions on your cover letter document which is attached.
